# Experimental investigation of orangutans' lithic percussive and sharp stone tool behaviours

Alba Motes-Rodrigo[1]*, Shannon P. McPherron[2], Will Archer[2,3], R. Adriana Hernandez-Aguilar[4,5], Claudio Tennie[1,2]

1 Department of Early Prehistory and Quaternary Ecology, University of Tübingen, Tübingen, Germany,
2 Department of Human Evolution, The Max Planck Institute for Evolutionary Anthropology, Leipzig, Germany, 3 Department of Archaeology and Anthropology, Max Planck Partner Group, National Museum, Bloemfontein, South Africa, 4 Department of Social Psychology and Quantitative Psychology, University of Barcelona, Barcelona, Spain, 5 Department of Biosciences, Centre for Ecological and Evolutionary Synthesis, University of Oslo, Oslo, Norway

* albamotes7@gmail.com

**Data Availability Statement:** All code, raw data, video samples and the 3D scan of the biggest sharp-edged stone produced by the orangutan are

## Abstract

Early stone tools, and in particular sharp stone tools, arguably represent one of the most important technological milestones in human evolution. The production and use of sharp stone tools significantly widened the ecological niche of our ancestors, allowing them to exploit novel food resources. However, despite their importance, it is still unclear how these early lithic technologies emerged and which behaviours served as stepping-stones for the development of systematic lithic production in our lineage. One approach to answer this question is to collect comparative data on the stone tool making and using abilities of our closest living relatives, the great apes, to reconstruct the potential stone-related behaviours of early hominins. To this end, we tested both the individual and the social learning abilities of five orangutans to make and use stone tools. Although the orangutans did not make sharp stone tools initially, three individuals spontaneously engaged in lithic percussion, and sharp stone pieces were produced under later experimental conditions. Furthermore, when provided with a human-made sharp stone, one orangutan spontaneously used it as a cutting tool. Contrary to previous experiments, social demonstrations did not considerably improve the stone tool making and using abilities of orangutans. Our study is the first to systematically investigate the stone tool making and using abilities of untrained, unenculturated orangutans showing that two proposed pre-requisites for the emergence of early lithic technologies–lithic percussion and the recognition of sharp-edged stones as cutting tools–are present in this species. We discuss the implications that ours and previous great ape stone tool experiments have for understanding the initial stages of lithic technologies in our lineage.

available in the OSF project link: https://osf.io/m9qtf/ or https://tinyurl.com/4rxbc9zb.

**Funding:** CT and AMR were funded by the European Research Council (ERC, https://erc.europa.eu) Starting grant awarded to CT (project STONECULT) under the European Union's Horizon 2020 research and innovation programme (grant agreement No. 714658). The funders had no role in study design, data collection and analysis, decision to publish, or preparation of the manuscript.

**Competing interests:** The authors have declared that no competing interests exist.

## Introduction

Given their resilience to destructive taphonomic processes, the most abundant hominin artefacts in the archaeological record are stone tools. Early stone tools typical of the Early Stone Age record include intentionally modified stones with sharp-edges (e.g. flakes and cores) and unmodified stone tools such as hammerstones and anvils dating as early as 3.3 Ma [1, but see 2–4]. The production and use of these and later stone tools is often highlighted as a milestone in human evolution: stone tools widened the early hominin ecological niche by enabling, for example, butchering [5], meat processing [6], bone marrow extraction [7] and plant tissue modification [8]. As a result, the production and use of stone tools elicited over time major changes in hominin dentition, hand morphology and brain size [9–11]. However, despite the clear ecological importance of stone tools in human evolution, it remains debated how the skills associated with their production and use emerged and how they were learned by naïve individuals.

Some of the oldest sharp stone tools excavated [e.g. from the Oldowan technocomplex; 12] show evidence of raw material selectivity [13–16] as well as technological skill underlying their production [16, 17]. These findings have led researchers to postulate potential evolutionary scenarios to explain the initial steps in the emergence of early stone technologies [18, 19]. One such scenario proposes that the production and use of sharp-edged stones was preceded by stone tool percussion techniques that involved the use of one or several unmodified stones [20–22]. These techniques are encompassed by the term lithic percussive technology [21]. The simplest lithic percussion technique involves striking or bashing a hand-held object (such as a fruit or a nut) against a fixed hard surface (generally a stone) that acts as an anvil [also called the anvil-only technique; 21, 23]. Another lithic percussion technique involves the use of a stone to forcefully strike on a solid body (e.g. a nut) placed on a hard surface (wood or stone) in order to access its contents [also known as hammer-and-anvil technique; 24]. Assuming the existence of such preceding techniques for the production of sharp-edged stones seems sensible given that several species of extant wild non-human primates (capuchin monkeys, *Sapajus* spp.; long-tailed macaques, *Macaca fascicularis*; and chimpanzees, *Pan troglodytes verus*) customarily use lithic percussion techniques to access a variety of food resources [reviewed by 24–29].

Given the extent of lithic percussion in the primate taxa and using cognitive cladistics, it has been hypothesized that the techniques employed to produce early sharp stone tools developed from–or were at least related to–lithic percussive behaviours similar to those exhibited by extant non-human primates [henceforth primates; 18–22]. However, reports of stone tool making (where a stone tool is produced for its subsequent use) are non-existent in wild primates. The only potential exception to this general observation was reported by Carvalho et al. [30], who among 1165 nut-cracking actions, observed nine instances where chimpanzee reused a fractured hammerstone that had broken during an earlier nut-cracking bout (although note that Carvalho et al. [30] considered these stone fractures unintentional).

The earliest hominin stone tool technologies are mainly based on the production and use of cutting tools [31]. However, stone-assisted cutting [in all its variants 32, 33] as well as the production of the corresponding tools, is absent in wild primates. A possible explanation why primates do not engage in stone-assisted cutting is that they can access and process food sources by other, more cost-efficient means [e.g. using their own teeth; 34]. Therefore, there may be no strong ecological need, and hence no (or little) evolutionary pressure for primates to develop and engage in the use of sharp stone tools for cutting [32, 35]. Nevertheless, the need to use sharp stones as cutting tools can be artificially created to experimentally determine if (and how) sharp stone tool production and use might develop in primates.

Two experiments in captivity have tested the abilities of great apes to make and use sharp stone tools when provided with food rewards that could only be accessed using cutting tools. The first experiment of this kind was conducted by Wright [36], who tested the abilities of a captive juvenile male orangutan, Abang, to use (Experiment 1) and make (Experiment 2) sharp stone tools to cut a rope-lock that allowed a baited puzzle box to be opened. During Experiment 1, Abang had access to the puzzle box and a sharp stone (a flake) produced by Wright out of sight of the orangutan. In Experiment 2, Abang had access to the puzzle box, a flint core with suitable knapping angles fixed on a platform and an unfixed hammerstone. During the course of these experiments, Wright performed numerous social demonstrations to Abang of how to use sharp stones as cutting tools (Experiment 1) and how to make sharp stones via freehand percussion (where a hand-held hammerstone is used to detach sharp flakes from a hand-held core, Experiment 2). Wright reports that at least on one occasion during Experiment 1, a keeper also molded Abang's actions guiding Abang's hand while cutting the rope lock of the puzzle box with a sharp stone. After nine demonstrations of how to open the puzzle box using a sharp stone as a cutting tool and 12 trials in which Abang had access to the testing materials, the juvenile orangutan used a (human-made) flake as a tool to cut the rope lock. In Experiment 2, after seven sessions including multiple demonstrations each of freehand percussion, Abang produced sharp stones himself by striking the fixed flint core with a hand-held hammerstone, a different technique from the one demonstrated by Wright given that the core was fixed during trials. Subsequently, Abang used (at least) one of the sharp stones that he had produced as a cutting tool to open the puzzle box.

A second project investigating the abilities of apes to make and use sharp stone tools was conducted by Toth, Schick and colleagues, starting in the 1990s. This project focused on the stone tool abilities of the enculturated bonobo Kanzi over a period of several years and across different experiments [37, 38]. The first set of tests was aimed at making Kanzi interested in sharp stones and their potential uses; the second experiment was aimed at training Kanzi to discriminate between stone artefacts of different sharpness; and the third set of tests was aimed at eliciting intentional sharp stone tool production. Before the onset of the experiments, Kanzi was provided with an undefined number of social demonstrations of how to make sharp stones via freehand percussion and use them to open a puzzle box similar to the one used by Wright [36]. After having been exposed to these demonstrations, Kanzi was given stones of different raw material types (quartzite, quartz, lava and chert) in forms that could be used as hammerstones and cores. In addition, Kanzi was allowed to manipulate the sharp stones produced by the human demonstrators and, occasionally, stones (hammers and cores) were placed in his hands to encourage him to actively participate in sharp stone production. In later experiments, Kanzi was also provided with a second drum-like puzzle box, which allowed access to food rewards more directly by using a sharp stone as a cutting tool to slit through a plastic cover.

Already on the first day of testing, Kanzi started to use the sharp-edge stones produced by the human demonstrators as cutting tools to access the baited box. In the second experiment, Kanzi was trained to identify the only sharp stone among five quartz stones that could be used to open the puzzle box. During the third experiment, Kanzi's abilities to make sharp stone tools himself were tested. Throughout the third experiment, Kanzi developed three sharp stone tool production techniques that had not been demonstrated to him by the human experimenters: repeatedly striking a stone resting on the ground with a hand-held stone (after 25 "experiences"); forcefully throwing a stone against a hard surface (in this case the floor, after 40 "experiences") and forcefully throwing a stone against another stationary stone lying on the ground [timing unspecified, 39]. During this same experiment, Kanzi also produced sharp stones using the freehand percussion technique that had been demonstrated to him. Later on, Kanzi's half-sister Panbanisha and his sons were reported to have learnt to knap, although no

details of the knapping techniques nor the learning process of these bonobos have been published [40, 41].

Although highly innovative at their time, these early experiments [36, 39] represent a limited contribution to our understanding of the origins of stone tool manufacture and use in our lineage for several reasons. First, neither of these studies explored the uninstructed innovation of sharp stone tool production and use by apes in a behavioural baseline, as demonstrations (and ocassionally molding) were provided to the apes. Consequently, it remains unclear which of the behaviours reported in those earlier studies were acquired from the demonstrations (by observational learning/molding), which were already in the apes' repertoire, and which could have developed spontaneously in the absence of guidance after having access to the testing materials. Given the large number of studies in the literature showing that great apes spontaneously express a wide range of tool behaviours in the absence of opportunities for observational learning [recently reviewed by 42], the possibility that apes could express sharp stone tool production and use on their own cannot be dismissed and should be tested [see also 43].

A second limitation of the earlier ape knapping studies is the assumption that the tested subjects were cognitively representative of their wild conspecifics (and by extension, informative regarding the cognitive capacities of the last common ancestor of these great ape species and hominins). This assumption is problematic given that all apes tested so far in knapping experiments were partly or fully enculturated. Enculturation refers to rearing conditions set "in a human cultural environment, with wide exposure to human artifacts and social/communicative interactions" [p.84, 44] that predispose the apes to interact with humans [45]. Enculturated apes also often undergo skill-specific training which can install cognitive skills [such as action copying; 46, 47] absent in unenculturated individuals [48–51] as so-called "cognitive gadgets" [52]. Indeed, Kanzi is perhaps one of the most extreme cases of enculturation, known for his extensive language training using lexigrams and his close contact with humans [53]. The orangutan tested by Wright [36], although less enculturated than Kanzi, was likely at least partly enculturated (e.g., Abang was taken for walks as a juvenile by his keeper and scientists entered the enclosure with the orangutan; John Partridge, former keeper of Abang at Bristol Zoo, UK, pers. comm.). Thus, given that enculturation can install cognitive abilities otherwise absent in wild-representative individuals, informative investigations using cognitive cladistics must test unenculturated subjects.

A third limitation of these early experiments is their sample size (N = 1 in both cases), which questions the generalizability of their results. Given these three limitations, further experiments using phylogenetically-representative great ape models are required to build data-informed hypotheses about the stone-related behaviours preceeding the emergence of lithic technologies in our lineage. Given the practical and ethical concerns associated to testing and provisioning wild individuals, we tested captive, task-naïve, unenculturated apes (specifically orangutans *Pongo pygmaeus*) in three experiments. In the first experiment we investigated the spontaneous abilities of orangutans to make and use sharp stone tools in the absence of demonstrations (Experiment 1). In addition to this behavioural baseline, we implemented a second experiment (Experiment 2) where we tested how increasing the value attributed to flakes influences orangutans' motivation to engage in sharp-stone making. Finally, we performed a conceptual replication of Wright's study (Experiment 3) with three unenculturated orangutans to evaluate the effect of social demonstrations on orangutans' sharp stone tool making and using abilities as well as the generalizability of Wright's results to other individuals [36].

We tested orangutans for several reasons. First, we intended to replicate and expand Wright's [36] study by testing unenculturated orangutans in the absence of copying (and molding) opportunities as well as after demonstrations. Second, orangutans are particularly interesting subjects of sharp stone tool making and using experiments because, despite being

proficient tool users and using a variety of raw materials as tools [54], they do not use stone tools in the wild. This absence of stone tool use behaviours in the wild orangutan repertoire further supports the naivety of our study subjects before the start of the experiments. Third, orangutans are the most arboreal ape species allowing us to build hypotheses about the potential stone tool abilities of hominid species with similar degrees of arboreality.

## Materials and methods

### Subjects and housing Experiments 1 and 2

Two male orangutans were tested in Experiments 1 and 2 at Kristiansand Zoo (Kristiansand, Norway). The two males lived in a family group including also a female and a new-born. As the new-born was still clinging to the female, the female orangutan often chose not to enter the testing quarters or interact with the testing materials, and consequently she was not the target of the experiments. The orangutan keepers confirmed the naivety of all individuals regarding stone tool making and use before the onset of the experiments and informed us that stones were absent from the outdoor enclosure of the orangutans because they are cleared regularly to prevent the orangutans from throwing them at the visitors. Both orangutans included in the experiment were mother-reared, lived in a group of conspecifics, only had direct contact with the keepers through the bars of the enclosure and had only been trained to allow veterinary procedures (a common practice in zoological institutions). No specific skill-training (e.g. language or action copying) had been conducted with these individuals. Therefore, we considered the orangutans at Kristiansand Zoo to be unenculturated and untrained and thus suitable for baseline testing.

The orangutans at Kristiansand Zoo have access to two enclosures (one indoor and one outdoor), as well as to a separate indoor sleeping area where the experiments took place. The outdoor enclosure is an island of approximately 1800 m$^2$ surrounded by a water-filled moat, with natural soil and artificial climbing structures. The indoor enclosure consists of several connected rooms up to 10 m high with a multitude of climbing structures and platforms. The indoor sleeping area consists of three rooms with two levels each. The sleeping rooms have concrete walls and floors as well as straw as bedding material. The orangutans have access to feeding enrichment such as tree trunks with holes filled with honey, which the orangutans obtain by using both their fingers and tools; automatic dispensers that release nuts into a maze, which the orangutans obtain using stick tools; and PVC tubes and hose fragments approximately 20 cm long with honey smeared inside. All orangutans participated voluntarily in the experiments as these took place during usual cleaning routines of the indoor enclosure. There were no changes made in their feeding routines during the course of the experiments, as the rewards used were part of their regular diet and water was available *ad libitum*.

### Testing materials Experiment 1

We provided both orangutans with two baited boxes that required the use of sharp stones as cutting tools: the tendon box and the hide box [see also 43]. Each box was baited with half a piece of fruit before each trial. The tendon box was modelled on the box described by Wright [36] and Toth et al. [39]. The tendon box was used to simulate a scenario in which, faced with an animal carcass, a subject must cut through taut tendons (a rope in our experiment) in order to dismember a body. The tendon box consisted of two boxes secured to a wooden board and a rope between them. The tendon box had a clear Plexiglas window (5 cm x 16 cm) at the top that allowed for the reward inside to be seen. The door of the first box was pulled shut by a rope that ran through the inside and exited through a hole in the opposite end. The rope then ran between the two boxes for approximately 5 cm and entered the second (non-rewarded)

box. Thus, the rope was only accessible in the area between the two boxes and had to be cut there to allow the door of the first box to open. The rope was a brown twisted hemp rope, approx. 1.5 mm thick.

The hide box was designed based on the second box used by Toth et al. [39] to test the stone tool making and using abilities of bonobos. The hide box was used to simulate a scenario in which, faced with an animal carcass, a subject must cut through taut skin/hide (a silicone membrane in our experiment) in order to access the inside of a body. The hide box consisted of a Plexiglass cylinder (16 cm wide x 15.5 cm high) with a metallic rim. A silicone membrane 2 mm thick was screwed in between the cylinder and the rim, blocking the access to the reward placed inside the cylinder. The hide box was then secured vertically to the bars of the sleeping rooms where the experiments took place.

One artificial hammer made of concrete (ca. 15 cm long x 10 cm in diameter, weight 2.2 kg) was provided during each trial. The hammer was built around a metallic scaffold linked to a chain that allowed us to fix the hammer to the bars of the sleeping rooms so the orangutans could not carry the hammers into the indoor enclosure (hence, the use of concrete). The concrete used to build the hammers included particles of up to 1 cm in diameter.

One prepared core of Norfolk chert was provided to the subjects alongside the hammer in each trial. The cores used in this study were prepared in advance to have edges suitable for flaking with angles varying between ~90 and ~35 degrees. The cores weighed between 0.8 and 1.5 kg. If the core was not modified during a trial, the core was used in further trials. Due to safety regulations, the core had to be fixed on a metallic platform (20 x 20 x 2 cm) to prevent the orangutans from carrying the core into the indoor enclosure (similar to the method used by Wright [36]). The core was attached to the platform using a metallic wired mesh from XTEND (Carl Stahl ARC GmbH, Architectural Cables and Mesh Systems) with a width of 50 mm and wire diameter of 3 mm.

In the second experimental condition implemented (*Use baseline*, see below for details) a human-made flint flake was additionally provided to the orangutans. The flake was made out of sight of the apes by the experimenter (AMR) using freehand percussion (see Introduction). The flake measured 7.6 cm in length, was 5 cm wide and had a maximum thickness of 1.7 cm. The flake was placed unfixed (loosely on the floor) next to the hammer, core and box before the subjects were allowed into the testing area.

## Set up Experiment 1

We tested subjects individually in two conditions in Experiment 1: the *Production and Use Baseline* and the *Use Baseline*. During the *Production and Use Baseline* the subjects were provided with the testing materials described above but no additional information (demonstrations, guidance, or stone artefacts) was provided. This first baseline was included in order to test for the spontaneous individual stone tool making abilities of the orangutans (production), and if tool making took place, for their spontaneous stone tool using abilities (hence the composite name of this condition). The fact that we did not provide any social demonstrations meant that the know-how (the physical actions) involved in sharp-edged stone tool making and use was not provided to the subjects. However, the experimental conditions implemented did not take place in an information vacuum. By introducing test materials into the testing quarters of the orangutans, we drew the individuals' attention towards a salient location where the materials were placed together ["local enhancement" or "know-where"; 55]. However, provisioning the testing materials did not convey any information regarding stone-related behaviours (techniques or know-how), making the *Production and Use Baseline* a valid test of reinnovation [55–58].

The *Production and Use Baseline* was split into two subconditions (*Production and Use Baseline* I and *Production and Use Baseline* II). During the *Production and Use Baseline* I, the two male orangutans had access to the tendon box, the hide box, a hammer and a fixed core. We included a second baseline (*Production and Use Baseline* II) in order to further focus the attention of the subjects by only providing them with one baited box instead of two. During the *Production and Use Baseline* II, the adult male received the tendon box (more robust than the hide box) and the juvenile was provided with the hide box (less robust). This decision was made based on our observations during the *Production and Use Baseline* I that the adult male could force open the hide box by hand.

The *Use Baseline* was identical to the *Production and Use Baseline* II except for the provision of a human-made chert flake together with the core, hammer and puzzle boxes (one per individual). Here, information regarding sharp stone tool making (a finished flake) was provided to the test subjects, making this condition a test of the spontaneous abilities of orangutans to use (and not produce) sharp-edged stone tools.

## Testing procedure Experiment 1

All testing materials (testing box/es, hammer and the fixed core) were placed on the floor inside the testing area and secured to the bars of the enclosure with metal wire and carabiners before the orangutans were allowed inside the rooms. The testing materials were spaced in a way that the orangutans could use the hammers on the core but not on the testing boxes. Two Sony HDR-CX330E Handycams were set-up half a meter from the bars and started recording once the subject entered the testing area. Potential tools were cleared from the testing areas before the tests started. However, the subjects often brought stick tools with them into the testing areas at the start of the tests.

Each orangutan was tested individually in three trials per condition. We planned for each trial of each condition to last 30 minutes. However, if the keepers decided that the animals showed signs of distress during any of the tests, the trial was immediately terminated. This occurred once while testing the juvenile, who was being trained to be separated from his mother at the time of testing. Furthermore, the duration of the trials had to be varied according to the cleaning routines, as this was the time when tests were conducted. Some keepers kept the doors to the outdoor enclosure open during trials so the orangutans could go in and out. In some cases (four trials), this led to especially short tests as the subject did not come back to the testing room. Trials in all conditions started when the subjects entered the testing room and ended when subjects exited the testing room for more than 10 minutes (in cases where the hatches connecting to the other enclosures were open, see below) or after 30 minutes, if the subjects stayed in the room.

The trials of the *Production and Use Baseline* I ranged in duration from 00:13:03 to 00:33:31; in the *Production and Use Baseline* II from 00:09:10 to 00:44:53 and in the *Use Baseline* from 00:11:03 to 00:55:36. The *Use Baseline* included four trials per individual due to the promising results obtained in the third trial of this condition.

Given the results of Experiment 1, we decided to conduct a follow up experiment in which we tried to elicit sharp-edged stone tool making by addressing some of the limitations that we thought might have affected the orangutans' performance in Experiment 1. The absence of sharp-edged stone tool making in Experiment 1 might have been caused by the orangutans not perceiving sharp-edged stones as interesting or valuable enough to make for themselves. Furthermore, the orangutans might not have identified the core as the source from where to detach sharp-edge stones. Therefore, the aims of Experiment 2 were to increase the value that the orangutans attributed to sharp-edged stones as well as to try to convey the association

between the core and the resulting sharp-edged stones. In other words, in Experiment 2 we provided additional information to the orangutans regarding the target know-what (sharp-edged stones) and target know-where (cores) but still without informing the orangutans about the target know-how (no demonstrations of any stone-related behaviours were given). Given that in Experiment 2 we introduced token exchanges with humans (a situation absent in the early hominins' and wild orangutans' environment), Experiment 2 did not simulate naturalistic conditions.

## Testing materials Experiment 2

The testing condition in Experiment 2 (*Flake Trading* condition) was preceded by a *Familiarization phase*. During the *Familiarization phase* we provided the orangutans with ten chert flakes that were produced in advance by the experimenter out of sight of the orangutans, using freehand percussion. During the *Flake Trading condition* of Experiment 2, the orangutans were provided with six of the ten flakes used in the *Familiarization phase*, a core and two artificial hammers equivalent to the one used during Experiment 1. In the *Flake Trading condition* the flakes were placed on top of the core to try to create a link between the core and the flakes (S1 Fig). The core provided to the orangutans in this condition had two detached flakes refitted into it. These flakes were made by the experimenter out of sight of the apes using freehand percussion. The idea behind placing these flakes (both the refitted ones and the non-refitted ones) on the core was not to convey information regarding how to obtain these flakes but to convey that flakes originated from the core. One of the flakes was weakly refitted to the core using a sugar-based glue (normally used for decorating cakes) and the other flake was strongly glued to the core with clear epoxy adhesive glue. Both glues were transparent and non-toxic (S1 Fig).

## Set up Experiment 2

We addressed the aims of Experiment 2 by testing the two orangutans individually in a *Familiarization phase* and a *Flake Trading condition*. In the *Familiarization phase* we intended the orangutans to exchange human-made flakes for food rewards in order for the orangutans to attribute value to flakes. After completing the *Familiarization phase*, the orangutans were then tested in the *Flake trading condition*. The aim of this experimental condition was to assess if, after the orangutans have directly associated flakes with high-value rewards (grapes), they would make sharp stones themselves in order to trade them and obtain more of these rewards. Two flakes were refitted to the core during the *Flake Trading condition* in order to increase the chances of success (flake detachment) if an orangutan hit the core with the hammer and to reinforce the association between the flakes and the core. We predicted that if a flake would detach easily, the orangutans would keep trying to make more, increasing the force used until it was sufficient to detach sharp stones from the provided core.

## Testing procedure Experiment 2

In the *Familiarization phase*, the human-made flakes were positioned on the floor of the testing room in two rows of five flakes each before each orangutan was allowed inside. Once an orangutan was allowed inside the room, the experimenter started counting 10 minutes using a stopwatch. As soon as the orangutan entered the room the experimenter asked the orangutan to trade a flake with her. The experimenter asked for flakes by showing the hand palm up, verbally encouraging the orangutan and occasionally pointing at the flakes [a usual procedure for ape-to-human token exchanges; e.g. 59]. If the orangutan pushed a flake out of the room, the experimenter gave him a grape. In order to consider the *Familiarization phase* successful, the apes had to exchange at least eight flakes in ten minutes. In the *Flake Trading condition*, loose

flakes (N = 6) were placed unattached around and on top the core structure to further reinforce the connection between flakes and core (see right panel of S1 Fig). The core and hammers were fixed as in Experiment 1. When an orangutan entered the room in the *Flake Trading condition*, he was immediately asked to exchange objects for grapes in the same way as in the *Familiarization phase*. Two Sony HDR-CX330E Handycams were set-up half a meter from the bars and started recording once the subject entered the testing area.

The orangutans were individually tested once in the *Familiarization phase* during 10 minutes and in four trials each during the *Flake Trading condition*, with each trial lasting approximately 30 minutes.

## Subjects and housing Experiment 3

Experiment 3 was conducted with a different population of orangutans housed at Twycross Zoo (Atherstone, UK). The decision to test a different orangutan group in Experiment 3 from the group tested in the previous experiments was taken in order to avoid carryover effects and to more accurately replicate the experimental experiences of the orangutan tested by Wright [36], who was not tested in a behavioural baseline. The orangutans included in Experiment 3 were three adult females, two of which had dependent offspring, who had no previous experience with the testing materials. Demographic data on the tested subjects can be found in S1 Table. The orangutans were housed in pairs (with their offspring) and all individuals had visual contact with each other and could interact through meshes. The composition of the pairs in which the orangutans were housed changed every two to three days. This housing arrangement was implemented in order to reduce the levels of aggression from the adult male in the group (not included in the study) towards the infants. The orangutans had access to indoor and outdoor enclosures as well as to off-sight quarters. The indoor enclosures were equipped with environmental enrichment such as climbing frames, bedding materials, platforms and containers where food can be placed for the apes to retrieve. The floor of the indoor enclosures was covered with wooden chips and straw. Outdoor enclosures consisted of grassed areas surrounded by glass walls. The outdoor enclosures included climbing frames and huts. Feedings took place several times a day, when food (fruit, vegetables, primate pellets and nuts) was scattered in the indoor and outdoor enclosures. Food was often placed inside enrichment devices such as hanging balls and boxes attached to the meshes. Water was available *ad libitum* in all enclosures.

The experiments took place in the off-sight quarters connected to the indoor enclosures. The experiments took place during cleaning routines, when the orangutans were individually housed in the off-sight quarters (mothers and dependent infants were tested together) and could not exit until cleaning routines in the enclosures finished.

## Testing materials Experiment 3

We used the same two puzzle boxes as in Experiment 1. Each box was baited before the onset of each trial with five peanuts or a small container with approximately 100 g of quick oats without shell soaked in water. These rewards were chosen based on the keepers' recommendations. The type of reward was chosen depending on the daily availability of each product.

Three artificial hammers made out of concrete were used during the demonstrations and the tests (small: 12 cm long x 9 cm in diameter, 2 kg; medium: 15 cm long x 10 cm in diameter, 2.5 kg; large: 18 cm long x 11 cm in diameter, 3 kg). The hammers had an overall potato shape and were built around a metallic scaffold linked to a chain that allowed fixing the hammer to a wooden platform. The concrete used to build the hammers included particles of up to 0.5 cm

in diameter. The concrete composition was altered from the previous experiments in order to strengthen the hammers and prevent their fracture after repeated impacts.

Chert cores equivalent to those used in the previous experiments were used for the demonstrations and tests. Different cores were used for the demonstrations and the tests of Experiment 3 (i.e. cores used during demonstrations were not used for testing). As in the previous experiments, the core was fixed on a metallic platform to prevent the apes from carrying the core into the indoor enclosure [similar to 36] or hitting the glass.

All testing materials in Experiment 3 were mounted onto two wooden platforms with a metallic frame that allowed us to fix the materials to the walls of the testing, off-sight quarters. In the first platform we fixed the core and the hammers while in the second platform we fixed the two testing boxes (S2 Fig). During tests, the platforms were fixed in opposite walls separated by 1.8 m to prevent the orangutans from hitting the boxes with the hammers.

## Set up Experiment 3

**Demonstrations.** The demonstrations to the orangutans were performed from a room connected to the testing room via a glass wall and to the indoor enclosure via a wall of rigid metal mesh (Fig 1). When demonstrations took place during cleaning hours of the enclosures, the orangutans could observe the demonstrations from inside the testing room through the glass wall (Fig 1C). When demonstrations took place outside of cleaning hours, these were directed to the orangutans in the indoor enclosure that could observe the demonstrations through the metal mesh. During the demonstrations, the orangutans could be as close as 1 m from the testing apparatuses. An individual was considered to have observed a demonstration when his/her head had been oriented with eyes open towards the demonstrator during the

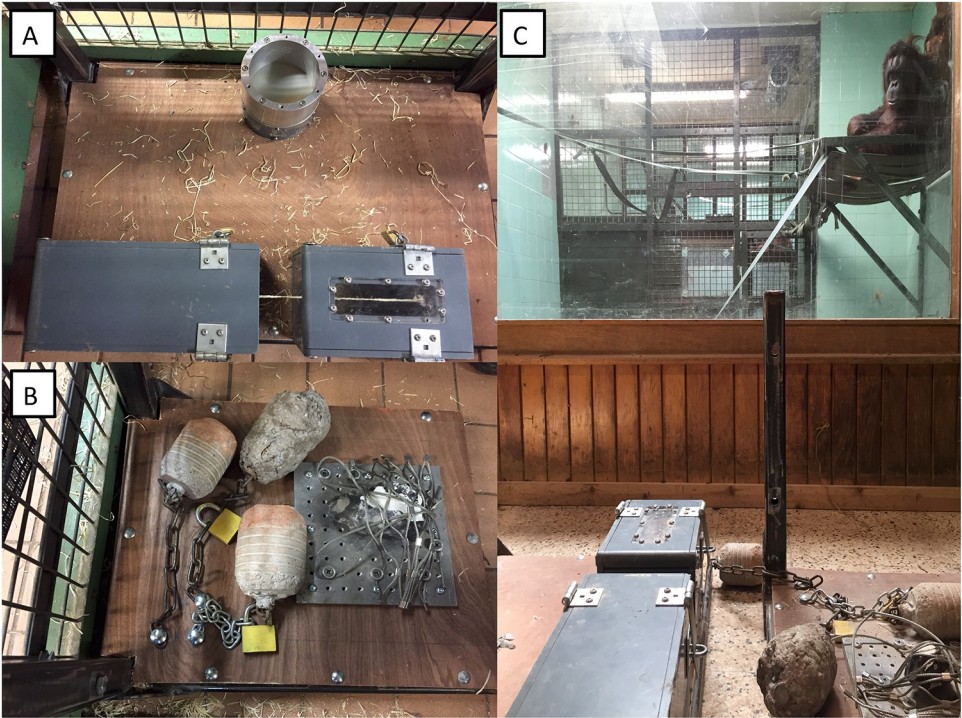

**Fig 1. Experimental set up.** Panel A depicts how the test boxes were fixed onto the wooden board and presented to the apes. Panel B depicts how the core and the three artificial hammers were fixed onto the wooden board and presented to the apes. Panel C depicts the room in which the demonstrations were given to the orangutans during cleaning routines.

entire demonstration. If the orangutan moved away or stopped looking during the demonstration, the demonstrator stopped and started again from the beginning when the orangutan was paying attention. If the experimenter was not sure if an individual had seen a full demonstration, the demonstration was repeated.

Each demonstration consisted of the production of one flake using a hand-held artificial hammer to strike a stabilized core placed on the fixing platform, the presentation of the flake to the orangutans and the subsequent use of the produced flake to open one of the boxes and obtain the food reward. The demonstrated knapping technique was chosen in order to show to the orangutans the production method that later was going to be available to them (contrary to Wright [36]). One flake was produced in each demonstration and flakes were not reused between demonstrations. After the detachment of a flake, the demonstrator held it in front of the apes to make sure that they saw it. The demonstrations of flake use did not start until the subject had seen the flake (i.e. their head was oriented towards the demonstrator). Demonstrations of flake use were conducted with both puzzle boxes (see below). When demonstrating how to open the tendon box, the demonstrator used the flake she had produced immediately before to cut the rope that kept the box closed. During the hide box demonstrations, the demonstrator used a flake she had produced immediately before to cut through a plastic sheet placed in the same position as the silicone membrane would be placed during the actual tests. We used plastic sheets instead of silicone membranes during the demonstrations due to the limited availability of the silicone membranes. When obtaining the reward, the demonstrator made sure that the ape saw it by taking the reward out of the box and showing it to the observing apes. After each demonstration, the boxes were re-baited with the same reward and closed.

The demonstrations involved all possible combinations (N = 9) between hand (left, right, both) and hammer type (small, medium, large) and were performed both with the tendon box and the hide box. Each of the nine combinations was demonstrated twice per test box (3 hand combinations x 3 hammers x 2 boxes x 2 rounds of demonstrations = 36 demonstrations) before the start of the test trials. Demonstrations were made to the orangutans in pairs (the pairs were set by the keepers according to their housing rotation routine, see above).

## Testing procedure Experiment 3

The young female orangutan Molly was tested individually whereas the adult females were tested with their dependent offspring. Tests and demonstrations focused on the adult orangutans because the infants were considered too young to participate in the tests. Each adult orangutan saw at least 27 demonstrations (three repetitions of the nine combinations) before the start of the tests (see Test calendar in OSF project folder for the individual delays between tests and demonstrations) and each adult was tested in four trials per condition (see below).

The orangutans participated in two experimental conditions equivalent to those reported in Experiment 1. In the first condition (*Social Production and Use*), individuals were provided with the testing materials after having been exposed to at least 27 demonstrations. In the second condition (*Social Use*), individuals were provided with a human-made flint flake alongside the testing materials. As before, the flake was made out of sight of the orangutans by the experimenter (AMR) using freehand percussion and was placed unfixed (loosely on the floor) next to the hammers and core before the subject was allowed into the testing area. Between the two experimental conditions, additional demonstrations (N = 18) were given to both adult females with offspring. These additional demonstrations were given to these two individuals because they did not show any promising behaviour in the initial tests, which could have been due to the long delay since they were exposed to the initial demonstrations. No additional demonstrations were given to the young female (Molly) between conditions due to the short delay since

she had seen the initial demonstrations as well as due to her motivation to interact with the testing materials.

Tests lasted between 23 and 60 minutes and a maximum of two tests were conducted per day (one in the morning and one in the afternoon). In one occasion and due to repair works in the indoor enclosure, one test lasted 90 minutes (the third trial of Mal in the *Social Use* condition). All demonstrations and tests were recorded with two Sony HDR-CX330E Handycams video cameras. A summary of the different conditions implemented can be found in Table 1.

## Coding

From each video-recorded trial of each condition we coded i) the number of interactions (events when the orangutans touched a testing material); ii) the duration of these interactions (time spent in physical contact with the testing materials, from the moment the subject started contact until it paused for more than 3s or changed activity); iii) which testing material the apes interacted with; iv) if the interaction was manual or using a tool; v) if percussion took place and vi) the number of strikes in a percussion event. As orangutans often use their mouths during tool use [60], oral interactions were also coded. In the *Familiarization phase* of Experiment 2 we coded the number of flakes exchanged in 10 minutes. Lithic percussive techniques could be defined based on the abovementioned variables.

A second coder recoded variables iii, v and vi from 110 interactions identified by the first coder. The interactions coded by the second coder where randomly selected using an online random number generator. Inter-rater reliability was calculated using the R function cohen. kappa from the package "psych" [61]. Kappa coefficients were 0.96, 0.87 and 0.74 respectively for variables iii, v and vi, which are considered very good agreement scores [62].

## Ethics

The experiments reported comply with the Guide for the Care and Use of Laboratory Animals (eighth edition, National Research Council, 2011), the American Society of Primatologists' Principles for the Ethical Treatment of Primates, and with current Norwegian laws. The experiments were approved by the Ethical commission of the European Research Council (ERC) and the ethical boards of Kristiansand Zoo and Twycross Zoo before their commencement.

## Results

### Experiment 1

None of the two orangutans tested detached sharp-edged stones from the core during the *Production and Use Baseline* I, II or the *Use Baseline*. However, both individuals frequently interacted with the testing materials (N = 907) across trials (S2 Fig).

**Table 1. Description of each of the conditions implemented in each experiment.**

| Experiment | Condition | N | Materials provided | Goal |
|---|---|---|---|---|
| 1 | Production and Use Baseline I | 2 | Puzzle boxes (2), hammer, core | Test spontaneous production and use |
| 1 | Production and Use Baseline II | 2 | Puzzle box (1), hammer, core | Test spontaneous production and use |
| 1 | Use Baseline | 2 | Puzzle box (1), hammer, core, human-made flake | Test spontaneous use |
| 2 | Flake exchange condition | 2 | Human-made flakes, hammer, core with refitted flakes | Test production |
| 3 | Social Production and Use | 3 | Puzzle boxes (2), hammers (3), core | Test production and use after demonstrations |
| 3 | Social Use | 3 | Puzzle boxes (2), hammers (3), core, human-made flake | Test use after demonstrations |

"N" represents the number of individuals tested in the condition. The "Materials provided" refer to the materials provided to each individual in each trial of a given condition. Puzzle boxes refer to the tendon and drum box described above. In the "Goal" column, "production" and "use" refer to sharp stone tools.

**Table 2. Frequencies, median durations in seconds and interquartile ranges (IQR, in brackets) of the interactions with the different testing elements by each orangutan.**

|  | Tendon box | Core | Hide box | Hammer | Total N |
|---|---|---|---|---|---|
| **Matthieu** | 247 (12; IQR = 18) | 9 (2; IQR = 5) | 16 (21; IQR = 42) | 12 (15; IQR = 13.5) | 284 |
| **Loui** | 107 (5; IQR = 7) | 24 (3; IQR = 3) | 397 (6; IQR = 9) | 95 (5; IQR = 10) | 623 |

Across all experimental conditions, the orangutans interacted the most with the baited puzzle boxes (S3 Fig, Table 2). When both baited puzzle boxes were available in the *Production and Use Baseline* I, the orangutans interacted more with the tendon box than with the hide box (S3 Fig). Interactions with the hammer increased from the first to the last condition as the orangutans started using the hammers to hit on the concrete walls and floor of the testing room, thus engaging in percussive behaviour. In addition, Matthieu, the adult orangutan, was able to break the chain tying the hammer to the room bars, which allowed him to manipulate the hammer much more freely and possibly led to a higher number of percussive events.

Interactions with the testing materials were made using the hands (N = 528) and the mouth (N = 166), as well as tools (see S2 Table). Different tools were used to interact with the testing materials, namely sticks (N = 9), a hose (N = 1) and the flake provided during the *Use Baseline* condition (N = 4).

**Tool use to open the boxes.** On the first trial of the *Production and Use Baseline* II, the juvenile orangutan Loui opened the hide box using a stick that he brought into the testing room. Using his body weight and exerting pressure with the stick, he succeeded in perforating the silicone membrane covering the hide box and obtained the reward. On the third trial of the *Use Baseline* condition, Loui tried to open the hide box using a hose fragment that he had brought from the indoor enclosure (these hose pieces were often provided to the orangutans as enrichment). After a failed attempt to open the hide box with the hose piece (which was not sharp), Loui proceeded to fetch the human-made flake from the floor, approximately 1 m away from where the hide box was fixed. Holding the flake with his mouth, Loui pressed the flake into the hide box membrane, piercing the membrane by pushing the flake and creating a hole that he then expanded by hand in order to obtain the reward (minute 00:24 of video in OSF). The other three uses of the flake (all performed by Loui) took place during the first trial of the *Use Baseline*: touching the core with the flake for three seconds; pressing the inside of the hide box with the flake for one second and pressing the outside of the hide box with the flake for four seconds.

## Experiment 2

During the *Familiarization phase* both subjects exchanged the ten human-made flakes provided by the experimenter for grapes within the set time limit of ten minutes (S3 Table). Consequently, both orangutans participated in the *Flake Trading condition*. During the test trials, the orangutans had the possibility to further exchange loose flakes placed around the fixed core for rewards (S3 Table) as well as the two refitted flakes glued to the core. The two orangutans exchanged with the experimenter both the refitted flakes and the loose flakes for food rewards during the trials. They obtained the refitted flakes by picking at them with their teeth and nails but not by using the hammer.

Percussive actions involving the use of the hammer as an active element to strike the walls and floor of the testing quarter were performed by both orangutans during the *Flake Trading condition* of Experiment 2 (Table 3). During two trials of the *Flake Trading condition*, the core

**Table 3. Ethogram of the behaviours performed by the orangutans when interacting with the hammers and core during Experiment 2.**

| Behaviour | Description | N | Median event duration (sec) | IQR |
| --- | --- | --- | --- | --- |
| Touch core | Subject places the hand on or touches the core. | 58 | 7 | 11.8 |
| Hit floor with core | Subject repeatedly strikes the core against the floor. | 7 | 5 | 5.5 |
| Hit wall with core | Subject repeatedly strikes the core against a wall. | 3 | 3 | 1 |
| Hit hammer on floor | Subject repeatedly strikes the hammer against the floor. | 33 | 4 | 3 |
| Interact with hammer | Subject places the hand on, touches or holds the hammer. | 77 | 9 | 8 |
| Hit hammer on wall | Subject repeatedly strikes the hammer against a wall. | 4 | 4 | 2.75 |

N refers to the count of behaviour frequencies and IQR refers to the interquartile range.

was extracted from the fixing platform by the juvenile orangutan Loui, which led to the performance of two previously unavailable behaviours (hit floor with core and hit wall with core).

**Sharp-edged stone detachment.** Loui, the juvenile orangutan, extracted the core from the fixed platform in the second and fourth trial of the *Flake Trading condition*. Once the core was loose, he proceeded to bash the core repeatedly and vertically against the floor (N = 7) and walls (N = 2) of the room (S4 Table, Fig 2) while holding the core with one hand.

On one trial, the bashing of the core led to the detachment of three sharp-edged stones (Fig 3). These pieces weighed 1.9, 0.6 and 0.3 g respectively. At the same time that these sharp-edged stones detached, the two refitted flakes (see Methods section) also detached. None of the detached sharp-edged stones (refitted or produced by Loui) were exchanged with the experimenter nor used for any other purpose and were abandoned together with the core once the orangutan was allowed outside the testing room.

## Experiment 3

The orangutans at Twycross Zoo interacted with the materials 425 times. Of these interactions, 48 took place using the mouth and the rest were manual interactions. The frequency of the interactions during Experiment 3 (S4 Fig) was highest during trial one of the *Social Production*

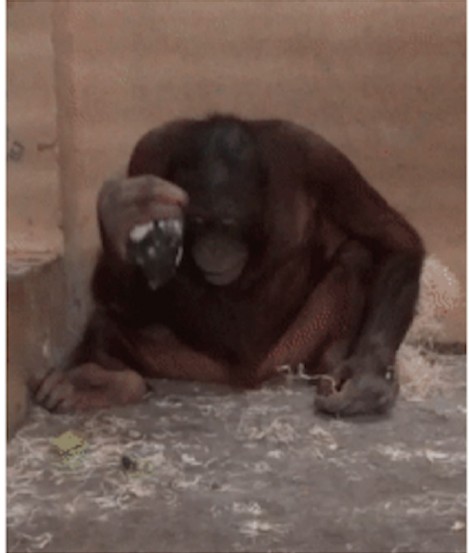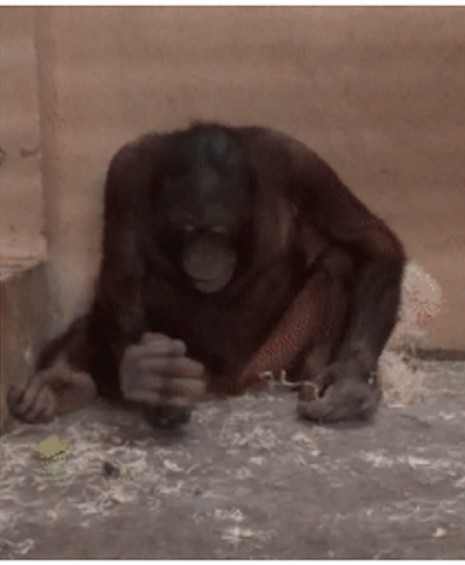

**Fig 2. Spontaneous lithic percussion.** Loui (the juvenile male orangutan) using the core as an active element to vertically strike on the concrete floor of the testing room during the *Flake Trading condition* of Experiment 2.

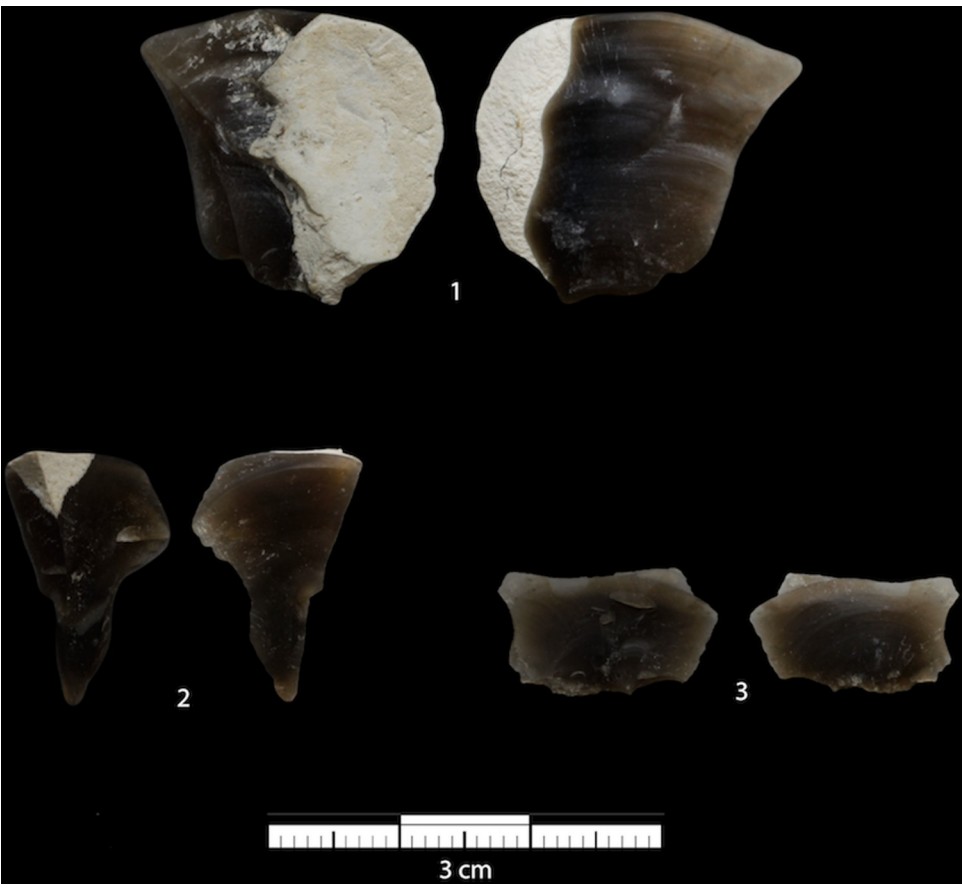

**Fig 3. Sharp-edged stones detached during percussive events by the juvenile orangutan in Experiment 2.** The heaviest piece (1) had a platform depth of 2.7 mm, a platform width of 6.8 mm, a width of 18.8 mm and a length of 20.3 mm. In the middle-sized piece (2), it was not possible to identify either the impact point or the platform. The maximum length and maximum width perpendicular to the length of the middle size piece were 19.05 mm and 11.36 mm respectively. The smallest piece (3) had an impact point, but no striking platform could be identified. The width of the smallest piece was 14.9 mm and the length 8.8 mm. Measurements were taken following the box method [63].

*and Use* condition (median number of interactions = 13, IQR = 51) and lowest during trial four of this same condition, when only three interactions took place. In addition, we also found substantial inter-individual variation with the younger female interacting more (median = 20, IQR = 20.8) with the testing materials than the two females with offspring (medians = 9.5 and 5, IQR = 25.2 and 6, S4 Fig).

Regarding the testing materials, the orangutans interacted the most with the tendon box (N = 129) and the hide box (N = 142, S5 Fig). Of the interactions with the tendon box, 23 involved touching the rope which needed to be cut in order to open the box and obtain the reward. The adult orangutans (as in Experiment 1) could manually open the hide box by ripping the silicone membrane. In an attempt to strengthen the hide box, the silicone membranes were substituted by two types of vinyl membranes (2 and 5 mm thick) and the attachments of the membrane were reinforced with metal rings. Unfortunately, the orangutans could still open the box after these modifications and therefore, although the hide box was baited and closed for every test, it was not operational. The orangutans interacted with the hammers a total of 76 times, with the core a total 21 times and during the *Social Use* condition, a total of 19 times with the human-made flake (S5 Fig, Table 4).

**Table 4. Individual number of interactions of each orangutan with the different testing materials.**

|  | Core | Hide box | Flake | Hammer | Tendon box | Total |
|---|---|---|---|---|---|---|
| **Kib** | 5 | 38 | 0 | 20 | 91 | 154 |
| **Mal** | 5 | 18 | 1 | 3 | 12 | 39 |
| **Molly** | 11 | 55 | 17 | 53 | 57 | 193 |
| **Total** | 22 | 137 | 18 | 82 | 168 | |

Although not the target of our observations, the two infants occasionally interacted with the testing materials (N = 45, N = 6 respectively), particularly with the open hide box (N = 23). However, their mothers often picked them up when they tried to play with the testing materials, thus preventing them from engaging in interactions longer than a couple of seconds.

Of the young female's interactions (Molly), 38 involved percussive actions in which an object was stricken against another. Most percussive actions were directed towards the floor, but also the wooden platform where the hammers and core were secured, the wall and the other hammers. In these cases, Molly held a hammer with one hand and repeatedly struck these surfaces, often changing the hand with which she held the hammer. These percussive actions with a (hand-held) hammer were performed both in a downward vertical motion and in a horizontal motion (when the wall was struck). As a consequence of these percussive actions, six pieces of concrete were detached from a hammer, which were licked and sniffed but not used in any way (Fig 4). Neither of the other two adult orangutans with offspring engaged in lithic percussion.

At least two out of the six pieces that detached as a by-product of the percussive actions had sharp enough edges that qualified them as potential cutting tools (which AMR tested later on by successfully cutting the rope of the tendon box with these hammer pieces out of sight of the orangutans).

Three times (once during the first trial of the *Social Production and Use* condition, Fig 5, and twice during the first trial of the *Social Use* condition), the young adult female (Molly)

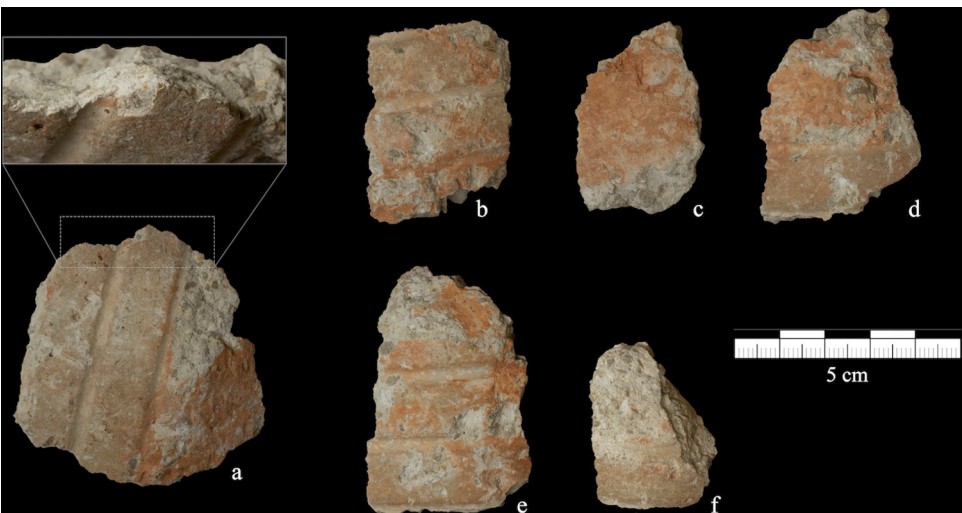

**Fig 4. Concrete pieces detached as a by-product of percussive actions.** These pieces were detached by the female orangutan Molly (9 years old) during events where a hand-held core (initially provided to act as a hammer) was repeatedly hit against a hard surface. Some sections of the pieces (enlarged in detail in the picture) were sharp enough to cut the rope closing the tendon box (tested by AMR).

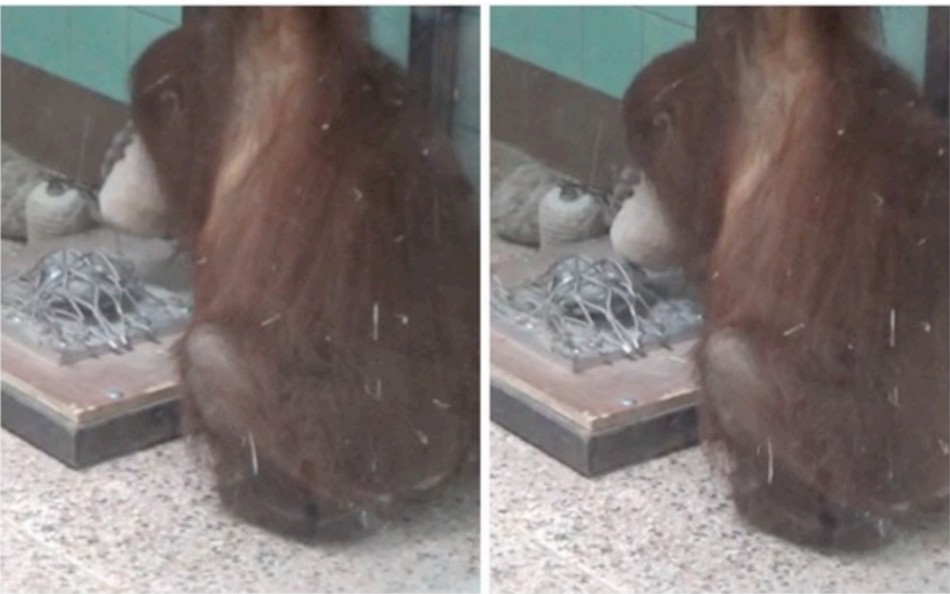

**Fig 5. Lithic percussion.** First percussive actions performed by Molly during trial one of the *Social Production and Use* condition using a hand-held hammer to strike the edge and exposed area of the core.

used a hammer to hit the fixed core. The first instance occurred 18 minutes after the start of trial one and lasted 14 seconds during which Molly hit the core 12 times holding the middle size hammer with the right hand. The second instance occurred 11 minutes after the start of the first trial of the *Social Use* condition. Molly struck the core three times in the span of seven seconds holding the larger hammer with the left hand. The third and last instance of percussion with a hammer on the core took place 38 seconds later, when Molly struck the core four times within ten seconds while holding the middle hammer with the right hand. Although no flake detachments took place, the strikes were directed to the exposed area of the core (one edge), from which flakes could have potentially been produced. This area is the one from which the flakes were detached during the demonstrations.

## Discussion

We conducted three experiments in which we investigated the stone-related behaviours of orangutans when provided with the necessary materials (cores and hammers) to make sharp stones and a motivation to use them as cutting tools (baited puzzle boxes). We explored the behavioural responses of both naïve orangutans and orangutans exposed to human demonstrations. In the *Produce and Use Baselines* of Experiment 1, neither of the two tested orangutans spontaneously produced sharp stones. However, both orangutans engaged in lithic percussive behaviours using the hammer provided. In the following *Use Baseline* we conducted a test of the spontaneous abilities of orangutans to indentify and use a (human-made) sharp stone as a cutting tool. In this condition, the juvenile orangutan (7 years old at the time of testing) used the provided human-made flake as a cutting tool to pierce the silicone closing the baited box.

Given the results of Experiment 1, we conducted a second experiment (Experiment 2) aimed at increasing the perceived value of sharp-edged stones by engaging the orangutans in flake-exchanges with a human. After several exchanges and a methodological mishap where the fixed core was removed from its fixing platform, the same juvenile orangutan that used a

flake in Experiment 1, now engaged in lithic percussion using the core as an active element to bash onto a hard surface (the concrete floor). This percussive behaviour led to the detachment of three sharp-edged stones, which were not exchanged for a reward (admitedly this behaviour only became available after the core's fixing system failed).

In Experiment 3, we replicated and expanded the study by Wright [36] testing the stone-tool making and using abilities of three orangutans housed at Twycross Zoo after these were exposed to human demonstrations. As in the previous two experiments, we observed lithic percussion using the hammers as active elements in one of the three subjects. However, contrary to our earlier experiments, some of these percussive events were directed towards the exposed area of the core from which sharp-stones could have been detached (know-where). In addition, and similar to Experiment 2, lithic percussion using the (moveable) hammers as active elements led to the detachment of hammer pieces (some of which were sharp).

## Lithic percussion

Wild orangutans have been reported to engage in organic percussion where they "hammer" termite or bee nests with wooden sticks that they previously modify in order to access insects or tough-skinned fruits [64]. However, no reports exist (to the best of our knowledge) of spontaneous lithic percussion in wild orangutans. Such lack of observations makes sense given that orangutans are the most arboreal great ape species and in the wild they are not often found on the ground [sensu 64]. Reports of orangutan lithic percussion in captivity were reviewed by Shumaker et al. [65] who described observations of single individuals at different zoological institutions using stones to "force locks, presumably by pounding on them" [66]; "pounding one stone with another" and "pounding on the glass with stones" (p114). Here, we show that both naïve orangutans (N = 2) and orangutans exposed to demonstrations (N = 1) engaged in lithic percussion. Given the absence of lithic percussion in wild orangutans, it may be that this behaviour is not expressed unless arboreality is partly substituted by terrestriality, as it occurs in captive settings [21, 67], and/or the right materials (e.g. mobile stones) are available. Interestingly, the number of percussive events did not increase substantially after social demonstrations were provided (47 spontaneous events vs 38 events after demonstrations). However, percussive actions directed to the core (know-where) were only observed in our study after human demonstrations had been provided. This result could indicate that although social demonstrations were not required for the orangutans to engage in lithic percussion, the demonstrations conveyed know-where information about the target of the percussive actions. In other words, the orangutan in Experiment 3 could have learned where to direct the percussive actions via local and/or stimulus enhancement. Our results add to previous studies on several other wild primate species [reviewed by 25] suggesting that lithic percussion might be deeply rooted in the primate lineage or alternatively, has emerged repeatedly throughout the primate phylogenetic tree [17, 68].

## Sharp stone production

Neither of the two orangutans that detached stone/concrete pieces in our study was directly trying to access food via percussive actions (i.e. no percussive actions could be directed to the boxes). Similarly, the young female in Experiment 3 did not use the sharp concrete fragments that she produced to try to access the food rewards in the tendon box. Instead, both individuals seemed to be playing or exploring the affordances of the hammers and core. Similarly, Shumaker et al. [65] report that "R. Wright's (1972) subject and a captive male (B. Beck pers. obs.) pounded with stones on cage walls and floors, both in play and in what seemed to be frustration" (p. 114). Percussive behaviours in non-foraging related contexts, such as object-directed

play, have been proposed to have an important role in the development of tool using skills both in modern human children [69] and chimpanzees [70]. Similarly to our results, an earlier primate study found that unenculturated tufted capuchin monkeys in captivity spontaneously engaged in lithic percussion in non-foraging contexts [58]. The results of these primate studies complement a growing body of literature on human children play [69, 71], suggesting that object-directed play and/or exploration might have had an important role in the development of stone tool using abilities in our hominin ancestors.

In Experiment 3 we failed to replicate the results by Wright [36] who found that after demonstrations, the juvenile male orangutan Abang made sharp-edged stones that he proceeded to use as tools. The lack of intentional stone tool making events in our study suggests that demonstrations alone are not sufficient to elicit successful sharp-stone making nor use in untrained and/or unenculturated orangutans. Contrary to Abang, the orangutans tested in our experiment were mother-reared, their behaviour was not shaped via molding and their direct contact with humans was limited to behavioural training for veterinary purposes. The rearing background of Abang, on the other hand, involved much more direct contact with humans from an early age and based on the reports obtained from the zoo where he was housed, Abang could be considered enculturated to some degree [44; J. Partridge pers. comm.]. Therefore, it is possible that demonstrations are not sufficient to elicit intentional sharp-stone tool making in unenculturated orangutans and that the degree of enculturation of ape subjects has a large influence in the development of these abilities.

Alternatively, although we attempted to use similar testing materials to those employed by Wright (equivalent testing box and provision of a fixed core), it is possible that differences in the core or hammer properties prevented the orangutan at Twycross Zoo from detaching sharp-edged stones during Experiment 3. First, due to safety regulations we could not use regular hammerstones as Wright did but had to create artificial hammers that could be chained to wooden frames. The lack of free moving hammers could have consequently limited the range of actions available to the orangutans, although we did observe the same core directed lithic percussion that in Wright's study led to the detachment of sharp-stones. Second, it is unclear which platform angles were present on the core Wright provided to the orangutan in his study. Differences in platform angle variability could partly explain the differences between Wright's results and our study as it is easier to detach sharp-stones from more acute platform angles [72]. Finally, it is likely that the lack of interactions with the testing materials of the two females that had dependent offspring was at least partially caused by the presence of the infants, as the mothers spent a considerable part of the trials trying to prevent the infants from interacting with the testing materials.

## Sharp stone tool use

Davidson and McGrew [34] pointed out that cutting has never been observed in wild apes and that only captive apes exposed to cutting demonstrations (as well as molding) had been shown to engage in this behaviour [36, 39]. Davidson and McGrew [34] argued that cutting, and the necessary identification of sharp stones for this purpose, had been a crucial prerequisite for the emergence of stone tool technologies containing modified stones. However, in Experiment 2, a juvenile unenculturated and untrained orangutan spontaneously identified and used a human-made sharp-edged stone as a cutting tool to perforate the silicone membrane of a baited puzzle box. Thus, our study presents the first evidence that unenculturated apes can also identify and use a sharp-edged stone as a cutting tool [at least for piercing; 33] in the absence of cutting demonstrations.

The orangutan that successfully used a human-made flake to cut through a hide-like structure held the flake with his mouth rather than with his hands (or feet). Such *oral tool use* might

not have been possible in Wright's study because the aperture of the baited box where the rope could be cut was too small for the ape to bring his lips close enough to the rope lock. Yet, this may be a preferred way of tool use for orangutans. Oral tool use is known to be common both in wild [73] and captive orangutans [60] as well as in other tool-using primates [64, 74]. Oral tool use, including oral sharp stone tool use, could have also played a role in lithic technologies of hominin species.

Our findings add to other primatological observations indicating that lithic percussion, the occasional (unintentional) detachment of sharp stones and the use of sharp stones as cutting tools (for piercing) might have been present in extinct primate species (including hominoids). Our findings suggest that the last common ancestor of *Pongo* and *Homo* may have also had the necessary cognitive and physical abilities to engage in lithic as well as organic percussive behaviour [21]. Given our relatively limited sample size, we hope that future studies further test this hypothesis including a larger sample of orangutans. In certain species of primates, such as baboons [75] and capuchins [76], immatures have been shown to engage more frequently than adults in object manipulation. If this would also be the case in orangutans, age differences might explain why it was the juvenile who innovated the use of a sharp-stone as a cutting tool in our study. Future work testing more orangutans from different age groups could assess potential age effects on the innovation and expression of lithic percussion and sharp stone tool use in this species. If extinct hominoids did engage in lithic percussive behaviour, it is possible that these individuals occasionally produced (perhaps unintentionally) sharp-edged stones during percussive foraging activities [as it is the case for chimpanzees and macaques; 30, 77] or even during object-directed play. However, finding these initial sharp-edged stones in the archaeological record may be challenging, especially if these artefacts were produced in low densities, not concentrated geographically and/or were morphologically similar to sharp-edged stones produced via non-anthropogenic processes [78]. Further studies with larger samples of orangutans would provide valuable information about the contexts (e.g. foraging or play) in which these behaviours are performed. Such studies would also provide reference collections to investigate the micro- and macroscopic use-wear signatures of stone detachment during orangutans' percussive activities, which could help identify products of lithic percussion potentially produced by orangutan-like hominoids.

## Conclusion

In the present study we report spontaneous lithic percussion as well as directed lithic percussion following demonstrations in captive, unenculturated orangutans. Lithic percussion led in two different occasions to the detachment of sharp stone/concrete pieces. Although the orangutans' performance did not change after observing social demonstrations in terms of behavioural form, lithic percussion directed to the core was only observed in one orangutan *after* social demonstrations had been provided. This result could indicate that the target of the percussive actions can be socially acquired in orangutans (perhaps via local enhancement). When presented with a human-made flake, a naïve orangutan spontaneously used it as a cutting tool to open a puzzle box, providing proof of concept that cutting (or piercing) using sharp-edged tools is within orangutans' spontaneous repertoire. Overall, our findings suggest that two prerequisites for the emergence of early lithic technologies–lithic percussion and the recognition of sharp-edged stones as cutting tools–might be deeply rooted in our evolutionary past (as old as 13 Ma). The differences between our results and the findings from previous ape knapping studies [36, 39] regarding the *association* of sharp-edge stone production and subsequent use as tools suggest that enculturation might be necessary for great apes to perform this complete behavioural sequence.

## Supporting information

**S1 Fig. Set up of Experiment 2.** The left panel illustrates the core provided during the *Flake Trading condition* together with the two refitted flakes shadowed in blue and red. The right panel illustrates the fixed core as presented to the apes, with the two refitted flakes and the loose flakes placed on top of the core.
(TIF)

**S2 Fig. Individual interactions during Experiment 1.** Number of interactions performed by each individual in each condition of Experiment 1. Blue dots represent the juvenile orangutan Loui and red dots represent the adult orangutan Matthieu. As there was no significant correlation between trial length and the number of interactions performed by the orangutans (Person correlation R = 0.016, p = 0.95), the results are displayed as sums of interactions rather than sums of interactions divided by trial length.
(TIF)

**S3 Fig. Interactions with testing materials during Experiment 1.** Number of interactions of orangutans towards the different testing materials in each experimental condition. Four of the interactions in the *Use Baseline* involved the human-made flake. In three occasions the flake was used to contact the drum and in one occasion to contact the core. The juvenile orangutan performed all the interactions involving the flake.
(TIF)

**S4 Fig. Individual interaction frequencies in each trial of Experiment 3.** Each color represents a different individual. Bold horizontal black lines represent median number of interactions across individuals. Boxes represent interquartile ranges (IQR).
(TIF)

**S5 Fig. Number of interactions with each of the testing materials in each trial of Experiment 3.** N represents the number of individuals tested in each trial.
(TIF)

**S1 Table. Demographic data of the orangutans included in the various experiments.**
(DOCX)

**S2 Table. Frequency and total durations of the interactions with the different testing materials, including whether the interactions involved a body part (hand or mouth) or a tool.**
(DOCX)

**S3 Table. Individual flake exchanges during the two conditions and total number of exchanged flakes.** In parenthesis is the proportion of flakes exchanged from those provided. "+2" indicates that the refitted flakes were also exchanged.
(DOCX)

**S4 Table. Duration, type and number of strikes during percussive events performed by the juvenile orangutan Loui with the core during Experiment 2.**
(DOCX)

## Acknowledgments

The authors are thankful to the staff at Kristiansand Zoo, especially Helene Axelsen and Tanya C. Minchin, for allowing us to conduct this study, and to the orangutan keepers for helping with the conduction of the experiments. The authors are also thankful to the staff at Twycross Zoo, especially to the ape team, for their help during data collection. AMR is also thankful to

Simon Fröhle for help with Figs 3 and 4 and to Prof. Maria Isabel Rodrigo Aleixandre for acting as second coder. The authors are also thankful to Prof. Susana Carvalho for helpful discussions on the topic of this manuscript.

## Author Contributions

**Conceptualization:** Alba Motes-Rodrigo, Shannon P. McPherron, Claudio Tennie.

**Data curation:** Alba Motes-Rodrigo.

**Formal analysis:** Alba Motes-Rodrigo.

**Funding acquisition:** Claudio Tennie.

**Investigation:** Alba Motes-Rodrigo.

**Methodology:** Alba Motes-Rodrigo, Shannon P. McPherron, Claudio Tennie.

**Project administration:** R. Adriana Hernandez-Aguilar, Claudio Tennie.

**Resources:** Will Archer, R. Adriana Hernandez-Aguilar, Claudio Tennie.

**Supervision:** Claudio Tennie.

**Visualization:** Alba Motes-Rodrigo.

**Writing – original draft:** Alba Motes-Rodrigo.

**Writing – review & editing:** Alba Motes-Rodrigo, Shannon P. McPherron, Will Archer, R. Adriana Hernandez-Aguilar, Claudio Tennie.

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
