## [Decision Letter · Decision Letter 0]

7 Dec 2021

PONE-D-21-30518Experimental investigation of orangutans' lithic percussive and sharp stone tool behavioursPLOS ONE Dear Dr. Motes Rodrigo,

Thank you for submitting your manuscript to PLOS ONE. After careful consideration, we feel that it has merit but does not fully meet PLOS ONE’s publication criteria as it currently stands. Therefore, we invite you to submit a revised version of the manuscript that addresses the points raised during the review process. The reviewers' comments are fairly straightforward and I would expect revisions to be easy to complete relatively quickly. 

We look forward to receiving your revised manuscript.

Kind regards,

Radu Iovita

Academic Editor

PLOS ONE

Journal Requirements:

2. We note that Figures 1, 5, Fig. S1, and S2, in your submission contain copyrighted images. All PLOS content is published under the Creative Commons Attribution License (CC BY 4.0), which means that the manuscript, images, and Supporting Information files will be freely available online, and any third party is permitted to access, download, copy, distribute, and use these materials in any way, even commercially, with proper attribution. For more information, see our copyright guidelines: http://journals.plos.org/plosone/s/licenses-and-copyright.

A. You may seek permission from the original copyright holder of  Figures 1, 5, Fig. S1, and S2 to publish the content specifically under the CC BY 4.0 license. 

B. If you are unable to obtain permission from the original copyright holder to publish these figures under the CC BY 4.0 license or if the copyright holder’s requirements are incompatible with the CC BY 4.0 license, please either i) remove the figure or ii) supply a replacement figure that complies with the CC BY 4.0 license. Please check copyright information on all replacement figures and update the figure caption with source information. If applicable, please specify in the figure caption text when a figure is similar but not identical to the original image and is therefore for illustrative purposes only.

3. We note that Figure includes an image of a participant in the study. 

Reviewers' comments:

Reviewer's Responses to Questions

**Comments to the Author**

1. Is the manuscript technically sound, and do the data support the conclusions?

Reviewer #1: Yes

Reviewer #2: Partly

2. Has the statistical analysis been performed appropriately and rigorously? 

Reviewer #1: N/A

Reviewer #2: Yes

3. Have the authors made all data underlying the findings in their manuscript fully available?

Reviewer #1: Yes

Reviewer #2: Yes

4. Is the manuscript presented in an intelligible fashion and written in standard English?

Reviewer #1: Yes

Reviewer #2: Yes

5. Review Comments to the Author

Reviewer #1: Overall, I enjoyed the simplicity and organization of the study. The authors are correct that we need additional observational studies of great apes’ capacities for lithic tool production and use. This is a valuable addition to the ape tool use literature and especially relevant for researchers interested in the origins of chipped stone tool production in early hominins.

Below, I include some more specific comments that I had on each section.

Introduction

There have been additional studies with the lithics-trained bonobos. I don’t think there’s much need to go on at great length about them, but they should at least be cited:

• Roffman et al. (2012). Stone tool production and utilization by bonobo-chimpanzees (Pan paniscus). PNAS 109(36),14500-14503.

• Roffman et al. (2015). Preparation and use of varied natural tools for extractive foraging by bonobos (Pan paniscus). American Journal of Biological Anthropology 158(1), 78-91.

All the limitations listed for previous ape toolmaking studies are valid.

Methods

I’m not sure that one can say that any ape that lives in a zoo is unenculturated. They are viewing tool-using people all day long everyday.

I would have hoped to see Experiment 1 conducted among wild apes with limited contact with researchers, but I can see how this would be extremely difficult to do, especially among such an arboreal, solitary species. I would like to at least see the authors acknowledge this limitation to their experiment design.

Were the materials for making a flake provided in the Use Baseline condition or not? This needs to be clearer.

While there are a lot of conditions to keep track of, I appreciated the organization of the experiment design. They covered all their bases quite well from what I can tell.

Results

I loved learning about the ingenuity of the apes (e.g., using a stick tool to perforate the membrane rather than a flake).

It might be useful to have a summary table of all the different conditions in the different experiments just as a reminder of the experiment setup for each condition while reading through the results.

What does “percussion length” refer to and how was it measured?

Discussion

I am wary of the conclusion that orangutans are unable to make sharp stone tools from watching demonstrations alone. There is literature that shows that ape imitation is socially directed. As Russon and Galdikas (1995:7) state, “…[imitation] follows kinship, affective relationship, and social status lines. The age of the ape influences whom it chooses to model. I am assuming that the apes in Experiment 3 were not very familiar with the experimenter who demonstrated stone tool production. So, it is not all that surprising to me that the apes did not model the demonstrated behavior. Once again, Russon and Galdikas (1995:15) state, “…experimenters have demonstrated totally novel actions to great ape subjects with models whose relationships with subjects were not clear…; the imitation they elicited was of low complexity. Our findings on selectivity predict such poor performance because such conditions may inhibit, not motivate, imitation in great apes.” I would like to see the conclusions toned down a bit in light of a more extensive discussion of this issue.

Reviewer #2: This paper investigates the lithic percussive and sharp stone tool behaviours in captive orangutans. The paper is interesting and I think it offers new comparative data on lithic production and use testing untrained, unenculturated subjects. Moreover, the experiments proposed to the orangutans are appropriate for the purpose to investigate individual and social learning abilities to make and use sharp stone tools.

I appreciate the effort to increase sample size in comparison to previous studies on the same topic but N = 2 or 3 are still a limited sample size and this obviously limits the external validity of the study. Moreover, using a species in which individuals do not use stone tools in the wild and use the mouth instead of the hands in performing many behaviours makes problematic to build hypotheses about the potential stone tool abilities of early hominins.

Subjects participated in 3 trials per condition in Exp. 1, 4 trials in the Flake trading condition in Exp. 2 and 4 trials per condition in Exp. 3. Why these number of trials? I wonder whether giving more trials to the subjects could have favoured the production/use of tools.

Results showed that subjects perform percussive behaviour with stone tools but it is not directed to “functional” parts of the apparatus presented. Thus although percussive behavior is present in their potential repertoire, orangutans do not seem to use percussive and sharp tools in a functional way (in this case to solve the tasks presented). I think authors should discuss these aspects more extensively giving possible explanations.

Sometimes in the text appears a “,” after a “;” (see for example Page 3 line 56, Page 3 line 66, Page 4 line 78). Please, remove the comma.

6. PLOS authors have the option to publish the peer review history of their article (what does this mean?). If published, this will include your full peer review and any attached files.

Reviewer #1: No

Reviewer #2: No

---

## [Author Response · Author response to Decision Letter 0]

7 Jan 2022

Journal Requirements:

––> We have now modified the manuscript formatting according to the instructions in the templates above. We have also changed the names of our figure and supplementary material files to follow journal requirements.

2. We note that Figures 1, 5, Fig. S1, and S2, in your submission contain copyrighted images. All PLOS content is published under the Creative Commons Attribution License (CC BY 4.0), which means that the manuscript, images, and Supporting Information files will be freely available online, and any third party is permitted to access, download, copy, distribute, and use these materials in any way, even commercially, with proper attribution. For more information, see our copyright guidelines: http://journals.plos.org/plosone/s/licenses-and-copyright. We require you to either (1) present written permission from the copyright holder to publish these figures specifically under the CC BY 4.0 license, or (2) remove the figures from your submission:

––> The images in the figures were taken by the authors and therefore are not copyrighted.

3. We note that Figure includes an image of a participant in the study. 

As per the PLOS ONE policy (http://journals.plos.org/plosone/s/submission-guidelines#loc-human-subjects-research) on papers that include identifying, or potentially identifying, information, the individual(s) or parent(s)/guardian(s) must be informed of the terms of the PLOS open-access (CC-BY) license and provide specific permission for publication of these details under the terms of this license. Please download the Consent Form for Publication in a PLOS Journal (http://journals.plos.org/plosone/s/file?id=8ce6/plos-consent-form-english.pdf). The signed consent form should not be submitted with the manuscript, but should be securely filed in the individual's case notes. Please amend the methods section and ethics statement of the manuscript to explicitly state that the patient/participant has provided consent for publication: “The individual in this manuscript has given written informed consent (as outlined in PLOS consent form) to publish these case details”. If you are unable to obtain consent from the subject of the photograph, you will need to remove the figure and any other textual identifying information or case descriptions for this individual.

––>We have modified the S1 Figure by removing the panel containing the picture of a zookeeper. Therefore S1 Figure does no longer contain identifying information of any human.

 We have checked our reference list for retractions using the Zotero plug in RetractionWatch and we have not detected that any of the works we cite in our manuscript have been retracted. We have also added the DOIs to each of the articles included in our reference list (when these were publicly available). We have also changed to italics all species names and corrected any typos we could find on journal abbreviations. Regarding other changes to the reference list, we have added references 12, 38, 63, 72, 75 and 76.

Reviewer #1:

 Overall, I enjoyed the simplicity and organization of the study. The authors are correct that we need additional observational studies of great apes’ capacities for lithic tool production and use. This is a valuable addition to the ape tool use literature and especially relevant for researchers interested in the origins of chipped stone tool production in early hominins. Below, I include some more specific comments that I had on each section.

––> We thank the Reviewer for this positive evaluation of our manuscript. Please see our responses to the Reviewer's comments below.

Introduction

There have been additional studies with the lithics-trained bonobos. I don’t think there’s much need to go on at great length about them, but they should at least be cited:

• Roffman et al. (2012). Stone tool production and utilization by bonobo-chimpanzees (Pan paniscus). PNAS 109(36),14500-14503.

• Roffman et al. (2015). Preparation and use of varied natural tools for extractive foraging by bonobos (Pan paniscus). American Journal of Biological Anthropology 158(1), 78-91. All the limitations listed for previous ape toolmaking studies are valid.

––> We thank the Reviewer for suggesting these references. We have included a reference to the stone tool study by Roffman et al. (2012) in line 178. Unfortunately, we could not find a place in the manuscript to appropriately cite the study by Roffman et al. (2015), given that we did not discuss tool excavation in the current manuscript. However, we are familiar with the study in question and have repeatedly cited it in one of our previous publications (e.g. Motes-Rodrigo et al. 2019 PLOS ONE).

Methods

I’m not sure that one can say that any ape that lives in a zoo is unenculturated. They are viewing tool-using people all day long everyday.

––>We agree with the Reviewer on the general point that measuring on a continuous scale the degree of enculturation of an ape living in captivity is not an easy task and that the dichotomic classification of individuals as enculturated/non-enculturated can miss potentially important information on the specific rearing conditions experienced by different apes at different institutions. However, in line with the literature, in the current manuscript we use the definition by Furlong et al. (2008), which states that enculturated apes are individuals that have been reared "in a human-cultural environment with wide exposure to human artefacts and social/communicative interactions" (Furlong et al. 2008, Anim Cogn, p. 84). We, like other researchers, interpret wide exposure to social/communicative interactions to mean that individuals participate in various kinds of human-specific rearing or skill training (such as language and action copying) (Subiaul 2016 Behav Sci). Therefore, according to this definition, the orangutans included in our study were not enculturated. To clarify our use of this definition we have included more information on this topic in lines 238-241.

I would have hoped to see Experiment 1 conducted among wild apes with limited contact with researchers, but I can see how this would be extremely difficult to do, especially among such an arboreal, solitary species. I would like to at least see the authors acknowledge this limitation to their experiment design.

––> We agree with the Reviewer that under ideal circumstances we would conduct the experiments presented here with wild apes to maximize ecological validity. However, food provisioning or even human-made object exposure is not encouraged in wild ape populations. In addition, as demonstrated by previous studies with wild and captive orangutans, wild individuals are extremely conservative and neophobic compared to captive individuals (Forss et al. 2015 Am J Primatol). Therefore, while it is theoretically possible to present puzzle boxes to wild orangutans, the feasability of such experiment and the amount of data that could be collected would be very limited. We have acknowledged this limitation in lines 255-256.

Were the materials for making a flake provided in the Use Baseline condition or not? This needs to be clearer.

––>We have now clarified in lines 465-467 that: " The Use Baseline was identical to the Production and Use Baseline II except for the provision of a human-made chert flake together with the core, hammer and puzzle boxes (one per individual)".

While there are a lot of conditions to keep track of, I appreciated the organization of the experiment design. They covered all their bases quite well from what I can tell.

––>We thank the Reviewer for this positive feedback.

Results

I loved learning about the ingenuity of the apes (e.g., using a stick tool to perforate the membrane rather than a flake). It might be useful to have a summary table of all the different conditions in the different experiments just as a reminder of the experiment setup for each condition while reading through the results.

––> We have now added a new table (Table 1) to the manuscript describing the goal of each condition implemented, the materials provided in each condition and the sample sizes tested.

What does “percussion length” refer to and how was it measured?

––>The percussion length of a flake is the maximum distance from the point of percussion to the distal end, following the axis of percussion (perpendicular to the striking platform width, Dogandžić et al. 2015 PLOS ONE). However, to facilitate readability, we have named this measurement "length" rather than "percussion length" (line 895). We have also specified in line 895 that we used the box method described by Debénath and Dibble (1994) to measure the stone fragments' length and width.

Discussion

I am wary of the conclusion that orangutans are unable to make sharp stone tools from watching demonstrations alone. There is literature that shows that ape imitation is socially directed. As Russon and Galdikas (1995:7) state, “…[imitation] follows kinship, affective relationship, and social status lines. The age of the ape influences whom it chooses to model. I am assuming that the apes in Experiment 3 were not very familiar with the experimenter who demonstrated stone tool production. So, it is not all that surprising to me that the apes did not model the demonstrated behavior. Once again, Russon and Galdikas (1995:15) state, “…experimenters have demonstrated totally novel actions to great ape subjects with models whose relationships with subjects were not clear…; the imitation they elicited was of low complexity. Our findings on selectivity predict such poor performance because such conditions may inhibit, not motivate, imitation in great apes.” I would like to see the conclusions toned down a bit in light of a more extensive discussion of this issue.

––> We thank the Reviewer for the comment and for pointing out these quotes from Russon and Galdikas (1995). We have now rephrased the sentence in line 1091 specifying that our results refer to unenculturated individuals without specific skill-training. Indeed, the results by Wright (1972), as well as other studies on enculturated apes, suggest that these individuals have developed imitative skills absent in their unenculturated counterparts (Buttelmann et al 2007 Dev Sci; Call 2001 Cybern Syst; Call and Carpenter 2003 J Educ Educ Ev). These skills are most likely the result of the process of enculturation itself (and especially targeted training), which can install human cognitive skills into apes intentionally or unintentionally (see also the comment by Tennie 2019 on Heyes 2018 Beh Brain Sci). Regarding the quoted article by Russon and Galdikas, there are several aspects that we would like to clarify. The definition of imitation used by Russon and Galdikas in their study is "incidents in which an observer replicates a model's behaviour and the replication is contingent on observing the modelling" (p.6). This definition is very broad and could readily refer to several social learning mechanisms such as contextual imitation via response facilitation or production imitation (i.e. copying of novel actions). Furthermore, Russon and Galdikas (1995) considered as imitation behavioural matching events whenever the "imitator" had not performed the target behaviour in the 10 minutes preceding the event. Consequently, these authors did not investigate production imitation (copying of novel behaviours), but rather other social learning mechanisms, in particular contextual imitation via response facilitation (compare Motes-Rodrigo et al. 2021 Royal Soc Open Sci; Byrne and Tanner 2006 Rev Int Psicol Ter Psicol; Call and Carpenter 2003 J Educ Educ Ev). Similar to other studies investigating social learning in apes, Russon and Galdikas (1995) study presents the additional limitation that no inter-rater reliability was conducted (see critique by Motes-Rodrigo et al. 2021 Royal Soc. Open Sci). In contrast, in the present study, the actions demonstrated to the orangutans were novel and not present in either the expressed or latent repertoire of the orangutans, as demonstrated by the results of the baseline conditions. Therefore, the comment of the Reviewer here refers to different learning mechanisms from those targeted by the present experiments (contextual versus production imitation). Finally, the extensive body of literature showing that unenculturated, untrained apes do not imitate novel behaviour (reviewed in Motes-Rodrigo and Tennie 2021 Biol Rev) further supports our conclusions that orangutans are unable to acquire sharp stone tools making skills from watching demonstrations.

Reviewer #2: 

This paper investigates the lithic percussive and sharp stone tool behaviours in captive orangutans. The paper is interesting and I think it offers new comparative data on lithic production and use testing untrained, unenculturated subjects. Moreover, the experiments proposed to the orangutans are appropriate for the purpose to investigate individual and social learning abilities to make and use sharp stone tools.

––> We thank the Reviewer for the positive evaluation of our manuscript.

I appreciate the effort to increase sample size in comparison to previous studies on the same topic but N = 2 or 3 are still a limited sample size and this obviously limits the external validity of the study. Moreover, using a species in which individuals do not use stone tools in the wild and use the mouth instead of the hands in performing many behaviours makes problematic to build hypotheses about the potential stone tool abilities of early hominins.

––> We agree with the Reviewer that despite our best efforts to test a larger sample of individuals compared to previous studies, our sample is still limited. We have now clearly acknowledged this limitation in lines 1162 and suggested further research avenues that would provide important information on different aspects of the initial stages of lithic technologies (lines 1162-1191). The first reason why we chose a species that does not spontaneously engage in stone tool use in the wild is because this ensured that the test subjects were truly task-naïve at the start of the experiments and therefore we could track the emergence of the target behaviours from the beginning. In addition, orangutans represent interesting models to evaluate the stone-related abilities that more arboreal and solitary hominoids might have presented (lines 304-313). Finally, we wanted to replicate the study by Wright (1972 Man) with unenculturated individuals to evaluate the generalizability of his results and the potential role that the enculturation process might have played in the development of the orangutan subject's skills.

Subjects participated in 3 trials per condition in Exp. 1, 4 trials in the Flake trading condition in Exp. 2 and 4 trials per condition in Exp. 3. Why these number of trials? I wonder whether giving more trials to the subjects could have favoured the production/use of tools.

––> Initially we planned to test each individual in 3 trials per condition in each experiment. The fact that we conducted 4 trials in experiments 2 and 3 was due to the promising results of the individuals in the previous trials of the respective experiments (lines 498-499). In the third trial of experiment 2, the juvenile orangutan engaged in lithic percussion that led to the detachment of sharp stone pieces. To test whether this individual orangutan would engage in further lithic percussion using the hammers as active elements to detach sharp stone pieces from a fixed core, we conducted a fourth trial. In experiment 3 we conducted four trials because the young female repeatedly engaged in lithic percussion, occasionally targeted towards the core. We conducted four trials because we wanted to assess whether, with more practice, she would use enough strength to detach sharp stone pieces from the fixed core using the hammers. The initial decision to test each individual in three trials was decided arbitrarily given that there were no previous studies testing the stone-tool making abilities of unenculturated apes that we could use as reference. In fact, there is scarce information in the literature regarding how long do great apes take on average to innovate novel tool behaviours and consequently, how long should innovation tests be. After completing the experiments reported here, we addressed this knowledge gap by conducting a literature review on the topic (Motes-Rodrigo et al. 2021 Am J Primatol). We found that great apes innovated most tool use behaviours within the first hour of testing, whereas each of our conditions was between 1.5 and 2 hours long. However, we agree with the Reviewer that it would have been interesting to test each individual for longer periods of time as the possibility remains that they could have expressed more behaviours in later testing trials.

Results showed that subjects perform percussive behaviour with stone tools but it is not directed to “functional” parts of the apparatus presented. Thus although percussive behavior is present in their potential repertoire, orangutans do not seem to use percussive and sharp tools in a functional way (in this case to solve the tasks presented). I think authors should discuss these aspects more extensively giving possible explanations.

––>Although it is true that we only observed one instance of an orangutan using a sharp edged stone as a cutting tool, this observations serves as proof of principle that orangutans have the ability to use sharp tools in a functional way. We have added in lines 1163-1169 a potential explanation (age effect) of why only a single individual used the sharp stones as tools to solve the task. These lines now read: " In certain species of primates, such as baboons [75] and capuchins [76], immatures have been shown to engage more frequently than adults in object manipulation. If this would also be the case in orangutans, age differences might explain why it was the juvenile who innovated the use of a sharp-stone as a cutting tool in our study. Future work testing more orangutans from different age groups could assess potential age effects on the innovation and expression of lithic percussion and sharp stone tool use in this species.". We have now also added in the Methods section in lines 474-476 and 652-653 that the orangutans could not have used the hammers as percussive tools to open the puzzle boxes because the testing materials were placed and fixed in such a way that the chain of the hammers was long enough to allow the orangutans to use the hammers on the core but too short to allow them to use the hammers on the puzzle boxes.

Sometimes in the text appears a “,” after a “;” (see for example Page 3 line 56, Page 3 line 66, Page 4 line 78). Please, remove the comma.

––> We have now corrected this typo throughout the manuscript.

---

## [Editor Report · Decision Letter 1]

17 Jan 2022

Experimental investigation of orangutans' lithic percussive and sharp stone tool behaviours

PONE-D-21-30518R1

Dear Dr. Motes Rodrigo,

We’re pleased to inform you that your manuscript has been judged scientifically suitable for publication and will be formally accepted for publication once it meets all outstanding technical requirements.

Kind regards,

Radu Iovita

Academic Editor

PLOS ONE
---

## [Editor Report · Acceptance letter]

21 Jan 2022

PONE-D-21-30518R1 

Experimental investigation of orangutans' lithic percussive and sharp stone tool behaviours 

Dear Dr. Motes-Rodrigo:

I'm pleased to inform you that your manuscript has been deemed suitable for publication in PLOS ONE. Congratulations! Your manuscript is now with our production department. 

Kind regards, 

on behalf of

Dr. Radu Iovita 

Academic Editor

PLOS ONE